# Enzymatic Hydrolysis of Kraft and Sulfite Pulps: What Is the Best Cellulosic Substrate for Industrial Saccharification?

Aleksandr R. Shevchenko [1], Ksenia A. Mayorova [1], Dmitry G. Chukhchin [1], Alexey V. Malkov [1], Evgeniy A. Toptunov [1], Vadim D. Telitsin [1,2], Aleksandra M. Rozhkova [3], Ivan N. Zorov [2,3], Maria A. Rodicheva [1], Vadim A. Plakhin [1], Denis A. Akishin [1], Daria N. Poshina [4], Margarita V. Semenova [3], Andrey S. Aksenov [1,*] and Arkady P. Sinitsyn [2,3]

[1] Northern (Arctic) Federal University, Northern Dvina Embankment 17, 163000 Arkhangelsk, Russia; a.shevchenko@narfu.ru (A.R.S.); ksu100103@yandex.ru (K.A.M.); dimatsch@mail.ru (D.G.C.); a.malkov@narfu.ru (A.V.M.); zhenyatope@gmail.com (E.A.T.); vadim.telitsin@gmail.com (V.D.T.); rodichevam@yandex.ru (M.A.R.); v.plahin@narfu.ru (V.A.P.); denis28_mbs@mail.ru (D.A.A.)

[2] Chemical Department, Lomonosov Moscow State University, Vorobyevy Gory, 1–11, 119992 Moscow, Russia; inzorov@mail.ru (I.N.Z.); apsinitsyn@gmail.com (A.P.S.)

[3] Federal Research Centre "Fundamentals of Biotechnology", Russian Academy of Sciences, Leninsky prospect, 33, build. 2, 119071 Moscow, Russia; a.rojkova@fbras.ru (A.M.R.); margs@mail.ru (M.V.S.)

[4] Institute of Macromolecular Compounds, Russian Academy of Sciences, Bolshoi VO 31, 199004 St. Petersburg, Russia; poschin@yandex.ru

* Correspondence: a.s.aksenov@narfu.ru; Tel.: +7-921-2915446

**Abstract:** Sulfite and kraft pulping are two principal methods of industrial delignification of wood. In recent decades, those have been considered as possibilities to pretreat recalcitrant wood lignocellulosics for the enzymatic hydrolysis of polysaccharides and the subsequent fermentation of obtained sugars to valuable bioproducts. Current work compares chemistry and technological features of two different cooking processes in the preparation of polysaccharide substrates for deep saccharification with *P. verruculosum* glycosyl hydrolases. Bleached kraft and sulfite pulps were subjected to hydrolysis with enzyme mixture of high xylanase, cellobiohydrolase, and β-glucosidase activities at a dosage of 10 FPU/g of dry pulp and fiber concentration of 2.5, 5, and 10%. HPLC was used to analyze soluble sugars after hydrolysis and additional acid inversion of oligomers to monosaccharides. Kraft pulp demonstrated higher pulp conversion after 48 h (74–99%), which mostly resulted from deep xylan hydrolysis. Sulfite-pulp hydrolysates, obtained in similar conditions due to higher hexose concentration (more than 50 g/L), had higher fermentability for industrial strains producing alcohols, microbial protein, or organic acids. Along with saccharification, enzymatic modification of non-hydrolyzed residues occurred, which led to decreased degree of polymerization and composition changes in two industrial pulps. As a result, crystallinity of kraft pulp increased by 1.3%, which opens possibilities for obtaining new types of cellulosic products in the pulp and paper industry. The high adaptability and controllability of enzymatic and fermentation processes creates prospects for the modernization of existing factories.

**Keywords:** wood lignocellulose; pretreatment; enzymatic saccharification; glucose; kraft pulping; sulfite delignification; biomodified pulp

## 1. Introduction

Food security principles around the world imply an increasing use of abundant non-starch polysaccharide sources, including wood, to produce important industrial bio-based products. That requires developing new technologies, including primarily efficient enzymatic hydrolysis of polysaccharides [1,2]. Deep saccharification is one of the main tasks in this context; it remains challenging due to the recalcitrant structure of wood lignocellulose [3,4]. Prior to the implementation of cost-effective techniques for cellulose and

hemicelluloses hydrolysis to fermentable sugars with advanced enzymes, an adequate pretreatment of raw substrate must be developed. A large number of researchers have proposed various pretreatment techniques based on physical [5,6], chemical [7], biological [8,9], and combined [10] treatment of wood lignocellulose. Nevertheless, the development of the most effective pretreatment method still remains urgent and continues to this day [11]. Finally, due to hierarchically ordered structure of cell wall and high lignin content in trees, pretreatment methods successfully applied in industry for non-wood lignocellulosics can hardly be adapted to pretreat raw wood for the production of sugars and subsequently bio-products with enzymes [12].

Novozhilov et al. first considered the industrial pulping of raw wood as a commercially available method for the effective pretreatment of wood polysaccharides to enzymatic hydrolysis [13,14]. The industrial wood-to-pulp processing provides leaching of most of lignin, partial destruction, and dissolution of hemicelluloses, as well as a reduction of cellulose degree of polymerization, increasing fiber swelling, etc. The widely applied sulfate or kraft pulping process implies reduced sulfur compounds and active alkali reagent and represents the dominant technology for production of paper-grade pulps from wood. It was shown that kraft pulping significantly increased the accessibility of hardwoods and softwoods polysaccharides for cellulases and hemicellulases [15–17]. Bleaching is one way of dealing with residual lignin as a negative factor for enzymatic saccharification. Bleached pulps have already been considered as an adequate substrate for enzymatic treatment [18]. Sulfite delignification was historically the first large-scale industrial pulping process [19]; it implies treatment with sulfur dioxide in acidic medium and at high temperatures for the production of commercial sulfite pulps of various applications. Enzymatic saccharification of sulfite pulps has been poorly studied so far; nevertheless, commercial cellulases have been applied in sulfite pulp production [20]. Various techniques have been proposed to modify sulfite pulping and produce sulfite pulps that are more suitable for enzymatic hydrolysis [21]. These include lignin conversion to lignosulfonates, as well as the application of the lignin reactive sites blocking agents to limit the enzyme adsorption onto lignin. An alternative approach is the enzymatic hydrolysis of industrially bleached pulps with minimal lignin content.

Successful pretreatment enables the complicated lignocellulose structure to be targeted by fungal glycosyl hydrolases (GHs). Generally GHs mixtures include endoglucanases, cellobiohydrolases, β-glucosidases, xylanases, and mannanases acting by endo- and exo-mechanisms, as well as accessory enzymes [22,23]. Acting together, these enzymes convert the main wood polysaccharides to fermentable sugars. However, simply maximizing the amount of carbohydrate active enzymes is not enough; the composition of the mixture and the ratio of individual enzymes are crucial for effective saccharification. The synergistic effect between cellulases and hemicellulases allows for a high saccharification level in cellulosic pulp with low enzyme dosages less than 10 filter paper units per 1 g of substrate. The generation of a cellulolytic complex with a balanced composition and high productivity is possible by means of genetic engineering. Currently, recombinant GHs produced by *Penicillium* fungi have received large-scale development and have demonstrated high efficiency in long-term studies [24,25] towards different cellulosic substrates [26], including good performance on kraft pulps [13,15,27]. In the 1990s, researchers of the Department of Chemical Enzymology of the Moscow State University started the development of laboratory-scale and industrial enzyme mixture derived from highly active *P. verruculosum* strains, which have already found applications as feed additives in animal breeding, increasing nutritional value [28,29]. The optimization of *P. verruculosum* enzyme compositions creates prospects for the successful saccharification of wood lignocellulose. The investigations of enzyme performance on less studied sulfite cellulosic pulp are now of great interest.

The enzymatic hydrolysis of cellulosic substrates produces fermentable sugars, glucose, xylose, mannose, and others; most of those can be converted into bioethanol [30] or other highly valuable products, such as organic or amino acids [31,32]. Another type of product is non-hydrolyzed residue, insoluble biomodified cellulose, which represents

a complex of partially hydrolyzed cellulose, xylan, and mannan. Due to the high hydrolysis level, its composition and properties differ essentially from the original, which could be favorable in terms of application [33,34]. Through comprehensive utilization of sugars, aromatic compounds, and insoluble products from lignocellulose, one can obtain the maximum benefit from non-food plant resources.

The current study presents a comparative analysis of enzymatic saccharification of two commercially available cellulosic pulps with *P. verruculosum* cellulases and xylanases—understudied sulfite pulp versus well-studied kraft pulp. The aim of the work was to validate existing industrial techniques for pulping and bleaching as effective pretreatment options of wood lignocelluloses and to select the optimal schemes for production of soluble fermentable sugars and valuable insoluble residues from wood polysaccharides.

## 2. Materials and Methods

### 2.1. Cellulosic Pulps

We used never-dried bleached spruce sulfite pulp and never-dried bleached hardwood kraft pulp (birch and aspen mixture 1:1). Sulfite and kraft pulps were commercially available and were produced according to the schemes presented in Figure S1 and Figure S2, respectively. The bleaching scheme for kraft pulp is presented in Figure S3. Commercial sulfite pulp was bleached in six stages using chlorine- and alkali-based agents. To determine the composition of raw pulps, an exhaustive hydrolysis (enzyme dosage of 30 FPU/g) was carried out with *P. verruculosum* enzymes according to [13]. The polysaccharide compositions of initial pulps (Table 1) were further calculated from the monosaccharide content of hydrolysates.

**Table 1.** Pulp composition.

| Component | Kraft Pulp, % | Sulfite Pulp, % |
|---|---|---|
| Cellulose | $72.0 \pm 1.4$ | $78.9 \pm 1.3$ |
| Xylan | $23.7 \pm 1.1$ | $7.0 \pm 0.5$ |
| Mannan | $3.0 \pm 0.6$ | $9.1 \pm 0.5$ |
| Others | $1.3 \pm 0.4$ | $5.0 \pm 1.5$ |

### 2.2. Enzymes

The enzyme mixture was produced by ascomycete *P. verruculosum* Xyl35/Xyl8 [35]. The enzyme dosage used in hydrolysis experiments was calculated from the specific cellulase activity on filter paper and was adjusted to 10 filter paper units (FPU) per 1 g of dry pulp. Corresponding activities towards carboxymethylcellulose (CMC), microcrystalline cellulose (MCC), beech xylan, galactomannan, p-NF-cellobioside (cellobiase activity), and p-NF-β-D-glucopyranoside (β-glucosidase activity) were determined and collected in Table 2. The activities were analyzed at a temperature of 50 °C and pH of 5.0, according to previously published techniques [36].

**Table 2.** Activities of the *P. verruculosum* enzyme complex (U adjusted to 10 FPU per 1 g of dry pulp).

| CMC-ase | MCC-ase | Xylanase | Mannanase | β-Glucosidase | Cellobiase |
|---|---|---|---|---|---|
| $286 \pm 1.94$ | $19.8 \pm 0.12$ | $2326 \pm 6.2$ | $29.1 \pm 0.15$ | $26.4 \pm 0.12$ | $10.9 \pm 0.08$ |

### 2.3. Enzymatic Hydrolysis of Pulp Substrates

Enzymatic saccharification at low pulp concentration (2.5 and 5% of dry fiber) was carried out using a ES-20/60 (BioSan, Latvia) laboratory shaker-incubator; for high pulp concentration (10%) a Biostat A Plus (Sartorius, Germany) bioreactor was used, all experiments were carried out in triplicates, and the results were presented as mean ± SD. The hydrolysis temperature was adjusted to 50 °C and pH was maintained at 5.0 with 0.05 M sodium acetate buffer. Continuous stirring at 150–300 rpm was applied during the entire

saccharification period, up to 48 h. Samples for analyses were taken after 3, 6, 10, 12, and 24 h (for 2.5 and 5% concentration) and after 1.5, 3.0, 4.5, 8, 12, 21, 24, and 48 h (for 10% concentration). The suspension was first centrifuged for 1 min at 13,400 rpm, and then the supernatant (hydrolysate) was further analyzed.

*2.4. Hydrolysate Analyses*

2.4.1. Determination of Reducing Sugars, Glucose, and Minor Soluble Sugars

An analysis of monosaccharides (glucose, xylose, and mannose), disaccharides (cellobiose, xylobiose) and oligosaccharides in hydrolysates was carried out by HPLC using an Agilent 1200 (Agilent Technologies, Santa Clara, CA, USA) high pressure chromatographic system according to a procedure such as [37]. Sugars were separated on a Dionex CarboPac PA20 (Dionex, Sunnyvale, CA, USA) column using NaOH gradient increasing from 7.5 to 100 mM and were analyzed using ESA Coulochem III (Thermo Fisher Scientific, Waltham, MA, USA) detector. The concentration of reducing sugars (RS, g/L) was determined by the modified Somogyi–Nelson method [38].

To assess the hydrolytic ability of the pulps, the oligosaccharides released under enzymatic action were further hydrolyzed using sulfuric acid. Hydrolysate aliquots were mixed with 8% sulfuric acid in a ratio of 1:1 and boiled for 20 min; after neutralization, the samples were analyzed similarly to the hydrolysates before acidic inversion.

The cellulose conversion in pulps was calculated as follows:

$$Cellulose\ conversion,\ \% = \frac{(Glc - 0.2 * Man) * 0.9}{Solids * \%\ cellulose} * 100\% \tag{1}$$

Here *Glc* is the concentration of glucose after inversion (g/L); *Solids* is the pulp fiber concentration for hydrolysis (g/L); *Man* is the concentration of mannose after inversion (g/L); *% cellulose* is the cellulose content in pulp according to Table 1; 0.2 is the conversion factor, taking into account the glucose monosaccharide produced from glucomannan (as mannose to glucose ratio in glucomannan is 4:1 according to literature data [39]); and 0.9 is a factor considering the addition of water molecules during hydrolysis.

2.4.2. Gravimetrical Analysis of Non-Hydrolyzed Residue

Pulp mass losses under hydrolysis were used to calculate total pulp conversion. Non-hydrolyzed residue, or biomodified pulp, was washed several times with distilled water to remove soluble products, and then it was frozen at −80 °C and freeze-dried using a Labconco FreeZone 2.5 machine (Labconco, Kansas City, MO, USA). The pulp conversion was determined by the formula:

$$Pulp\ conversion,\ \% = \left(1 - \frac{non-hydrolysed\ residue\ dry\ weight\ (g)}{initial\ fiber\ dry\ weight\ (g)}\right) * 100\% \tag{2}$$

*2.5. Pulp Fiber Characterisation prior and after Enzymatic Hydrolysis*

To determine fiber dimensions, original never-dried pulps were suspended in water to a dry fiber concentration of 0.5% and analyzed using L&W Fiber Tester. Objects less than 0.1 mm wide and more than 0.2 mm long were counted as fibers [15].

The degree of polymerization (DP) of cellulose in initial pulps and lyophilized biomodified residue was calculated from intrinsic viscosities of 0.1% cellulose solutions in cadmium ethylenediamine according to Shevchenko et al. [27]. Pulp crystallinity was determined according to the previously published method [40] using a XRD-7000S (Shimadzu, Japan) powder X-ray diffractometer. The X-ray diffractograms of pressed pellets were recorded using a non-reflective silicon sample holder and X-ray tube with Cu target that operated at 50 kV and 30 mA.

## 3. Results

### 3.1. Enzymatic Hydrolysis at Low Pulp Concentrations

We compared cellulose conversion in sulfite and kraft pulp, taking into account differences in their polysaccharide composition. The saccharification was performed at fiber concentrations of 2.5 and 5.0% (*w/w*, dry weight) with 10 FPU/g of *P. verruculosum* enzymes for 24 h. The concentration of 2.5% was chosen to ensure complete hydrolysis. The concentration of 5% was chosen to increase sugar concentrations in hydrolysates while hydrolysis levels were still high. Kraft pulp showed a high level of cellulose conversion up to 93% (Figure 1). Sulfite pulp demonstrated a lower level of cellulose conversion; however, 50% of the theoretical glucose yield was reached in 12 h (Figure 1, concentration 2.5%). Increasing the substrate concentration from 2.5 to 5% reduced cellulose conversion for both pulps, with the largest decrease of 33% for kraft pulp, which is related to a decrease in stirring efficiency and the presence of residual lignin causing enzyme inhibition [41]. The total amount of reducing sugars over 48 h was at the level of 27.2–51.9 g/L for both pulps (Table S1). The maximal pulp conversion was achieved at the lowest substrate concentration (2.5%), 99% and 94% for kraft and sulfite pulp, respectively (Table S1). Thus, at low substrate concentration, both pulps demonstrated high hydrolyzability since they were almost completely hydrolyzed by the *P. verruculosum* enzyme complex.

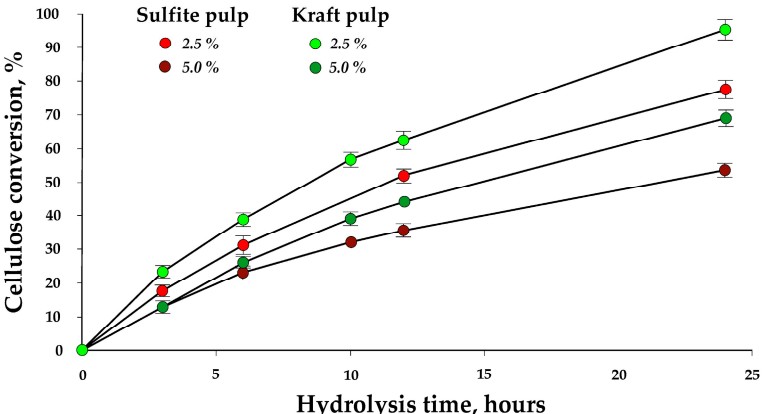

**Figure 1.** Cellulose conversion by *P. verruculosum* enzymes at low pulp concentrations.

### 3.2. Enzymatic Hydrolysis at High Pulp Concentrations

Enzymatic hydrolysis of lignocellulosics at high substrate concentration of 10% will provide higher concentrations of sugars in resulting hydrolysates, which is crucial for subsequent biosynthesis of organic acids and amino acids. However, higher pulp concentrations require more intensive stirring. In that case, several GHs can be inhibited by corresponding end products. We also observed a significant decrease in the hydrolysis efficiency, by 10–30%, with an increase of the pulp concentration from 5 to 10%. Hydrolysates of kraft and sulfite pulp differ in the composition and content of soluble sugars. As shown in Figure 2, glucose, being a cellulose monomer, dominated in hydrolysates over the course of hydrolysis. The final glucose concentration after 48 h reached 50 g/L for sulfite pulp, while 4% less glucose concentration was obtained from the kraft sample under the same condition. The hydrolysates differed most significantly in content of xylose, a xylan monomer. After 48 h, the ratio of xylose to glucose was 0.35 for kraft pulp and only 0.06 for sulfite pulp. In addition, the concentration of reducing sugars was also different for two types of hydrolysates, reflecting the mannan content in two pulps. For kraft pulp, the concentration of reducing sugars was 71.1 g/L at a total pulp conversion of 74% (Table S1). The final conversion of sulfite pulp was lower at 70%; however, sulfite pulp produced glucose-rich hydrolysates, and minor sugars represented less than 18% of hydrolysate monosaccharides.

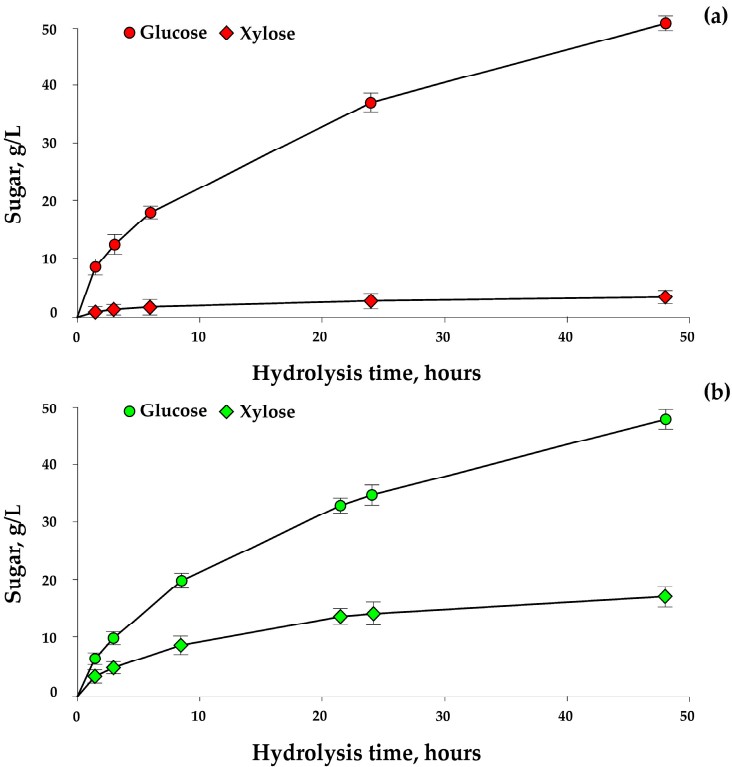

**Figure 2.** Accumulation of glucose and xylose during enzymatic hydrolysis of industrial sulfite (**a**) and kraft (**b**) pulps at high pulp concentration.

*3.3. Transformation of Non-Hydrolysed Residue*

As hydrolysis proceeded deeper with the accumulation of large amounts of soluble sugars, significant changes occurred in the composition and properties of insoluble residue. Such biomodified pulps consist of low DP cellulose, remaining hemicelluloses, and non-carbohydrate compounds [27]. As shown in Figure 3, the initial DP of cellulose in pulps after sulfite delignification was higher compared to kraft pulp—1025 ± 122 versus 950 ± 92, respectively. After hydrolysis with *P. verruculosum* enzymes, the average length of cellulose chain significantly decreased in 24 h and further; however, this modification proceeded differently for two pulps. While the DP for sulfite pulp decreased almost twice in 48 h (DP 550), at the same time for kraft pulp at a conversion level of 74% the DP reduced more drastically and reached the level of commercial microcrystalline cellulose (DP 290).

Crystallinity is another important parameter that is crucial for possible further applications of biomodified pulps. As shown in Figure 4, the initial values of the crystallinity for two pulps were quite similar and equaled 46–47%. At the initial stages of hydrolysis, up to 24 h, crystallinity of commercial pulps generally increases due to the action of enzymes, especially hemicellulases and endo-β-1,4-glucanases, which predominantly digest amorphous cellulose [27]. We also observed an increase in crystallinity of 1.3–2.2% in both pulps after 24 h of hydrolysis. The resulting crystallinity was slightly higher than the previously obtained values for commercial microcrystalline cellulose (47.2%) [27,33]. The final hydrolysis stages (48 h) contributed to a decrease in the crystalline areas, with more significant changes for sulfite pulp (Figure 4). The crystallinity of kraft pulp after prolonged hydrolysis remained at a high level, higher than that of initial kraft pulp and commercial microcrystalline cellulose.

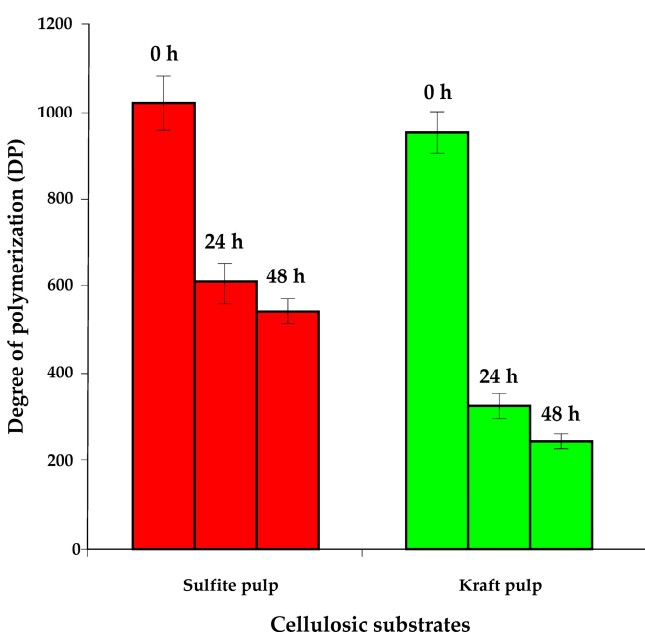

**Figure 3.** Degree of polymerization for initial and biomodified pulps.

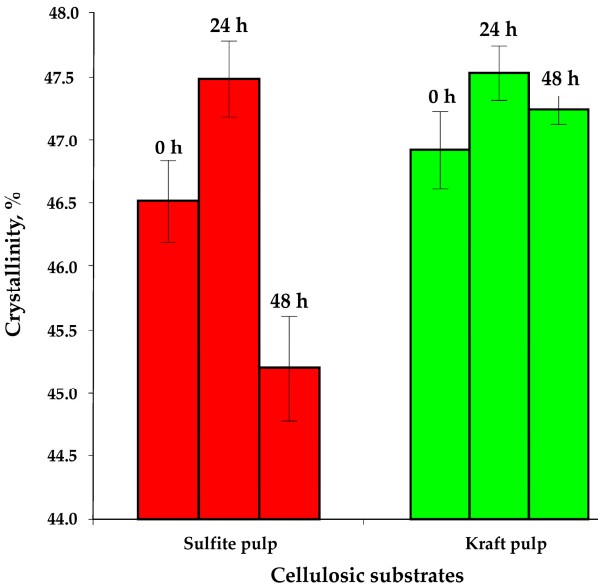

**Figure 4.** Crystallinity changes in cellulosic pulps under hydrolysis by *P. verruculosum* enzymes.

## 4. Discussion

### 4.1. Feasibility of Industrial Pulping for Enzymatic Hydrolysis of Wood Polysaccharides

Wood lignocellulosics contain a significant amount of lignin, an aromatic heteropolymer, that binds polysaccharides into a complex matrix and in the meantime hinders the cellulase attack on the cellulose. The enzyme inactivation occurs during the non-productive binding of cellulases on lignin due to electrostatic or hydrophobic interactions [15,41,42]. Enzyme inhibition with an excessive amount of acetyl groups or some extractives can also contribute to a decrease in the hydrolysis efficiency [43]. Thus, for successful enzymatic saccharification of wood lignocellulosics, it is necessary to minimize the factors during the pretreatment stage. On a laboratory scale several methods are used to delignify wood, such as steam explosion [44], treatment with mineral acids [45] or organic acids [46], sodium hydroxide [46,47], and organosolv pulping [48]. Despite the high efficiency of these methods, technological and economic difficulties arise when scaling them up. An

industrially available alternative is the thermochemical processing of wood chips aimed at obtaining fibrous wood pulps for the manufacturing of paper, cardboard, and other paper products. Enzymes that have already found an application in the pulp and paper industry include endoglucanases for pulp viscosity control and xylanases for kraft pulp prebleaching [20]. This creates a background for a deeper introduction of enzymatic steps into the production process.

Commercial pulps contain more than 92% of polysaccharides, consisting of cellulose as the major component and hemicelluloses (up to 30%) that were partially destructed during cooking. Compared to original wood, commercial pulps have lower lignin content, improved accessibility to cellulases and hemicellulases, and a lower content of fermentation inhibitors after hydrolysis. The toxic inhibitors are mostly products of monosaccharide thermal degradation, furfural and 5-hydroxymethylfurfural, phenolic compounds derived from lignin, and organic acids, which can cause significant inhibition of industrial microbial strains [49]. Since these compounds are concentrated mainly in spent cooking liquors, the hot washing of commercial pulps removes most of the inhibitors from the fibers. Furthermore, alkaline treatment stages using sodium, calcium, or ammonium hydroxide, at a pH higher than 10, significantly enhances the detoxification of cellulosic pulps [50].

Historically, sulfite pulping was developed earlier than other methods of industrial delignification [51]. The most common process is acid sulfite cooking (Figure S1), which applies acidic cooking liquor produced by burning sulfur or pyrites. Under the action of sulfur dioxide in the cooking liquor, lignin is transferred into a solution in the form of lignosulfonates [52]. High temperatures and the action of sulfurous acid and its salts lead to an efficient extraction of lignin, partial destruction of hemicelluloses and, to a lesser extent, cellulose. As a result, cellulose-rich pulps are produced which contain less than 20% mannan and xylan in total (Table 1). Sulfite spent liquor, a by-product of sulfite pulping, contains a mixture of oligo-, di-, and monosaccharides with a total content of 26–30% of the dry weight, as well as lignin derivatives and sugar decomposition products [53]. Further treatment of spent liquor at sulfite plants includes neutralization, evaporation, and other chemical and thermomechanical treatment, leading to liquor detoxification. For more than 85 years, spent liquor sugars have been utilized by yeast fermentation to obtain feed protein or ethanol [54]. Among industrial strains, yeasts are the most resistant to the highly inhibitory liquor-based media, but poor substrate quality results in low fermentation productivity and low overall technology efficiency.

To date, kraft pulping is one widespread and straightforward way to obtain cellulosic pulps. It involves treatment with a sodium hydroxide and sodium sulfide (Figure S2) and ensures the extraction of most of the lignin, acetyl groups, uronic acids, and partially hemi-celluloses [55,56]. The disruption of lignin–carbohydrate linkages during kraft pulping results in the subsequent liberation of fibers with an increase in their inner surface area. This and subsequent bleaching (Figure S3) positively affect the pulp papermaking properties, and also determines the high accessibility of kraft pulps to enzymatic hydrolysis, according to present and previous studies [15]. A well-designed recovery system for cooking chemicals and heat eliminates lignin utilization issues. Spent black liquor contains 30–35% lignin and 30–35% sugar decomposition products; its combustion provides energy to heat the digester. The green liquor produced after combustion is subjected to causticization to obtain white liquor, an active reagent for the cooking of a new batch of wood chips (Figure S2).

Both sulfite and kraft pulping produce fibrous paper-grade pulps, however, the result-ing pulps differ significantly in fiber properties and composition, which directly depends on the cooking process, cooking reagents, and on the species of wood [13]. During sulfite pulping, polysaccharides, mainly hemicelluloses, are partially degraded and released into the solution in the form of dextrins, oligo-, and, to a lesser extent, monosaccharides [57]. The acidic environment causes deacetylation of hemicelluloses; deacetylation and cleavage of galactopyranose units in softwood glucomannans lead to their consolidation, which, together with the high resistance of cellulose to cooking acid, contributes to an increased

yield of sulfite process comparing to kraft pulping. For these reasons softwoods, containing mainly cellulose and mannans, are used to obtain high pulp yields. Alkaline delignification is characterized by a higher loss of cellulose, which is less resistant to alkaline peeling compared to hemicelluloses. Hardwoods are considered the most suitable for kraft pulping, mainly due to the unique behavior of xylans during cooking. In an alkaline medium, most of the xylans dissolve, however, after the cleavage of glucuronic acid units, xylan quickly stabilizes on cellulose fibers, which increases both the mechanical strength and the yield of the kraft fibers. Such modified xylan becomes more accessible for hemicellulases; we observed its deep hydrolysis under the action of the *P. verruculosum* xylanases. Since xylan destruction made pulp fibers more accessible to cellulases, the close to complete saccharification of hardwood kraft pulp became possible due to the synergistic action of cellulases and hemicellulases in the enzyme mixture. As a result, we obtained a hydrolysate with a predominantly higher content of monosaccharides (glucose and xylose) compared to oligosaccharides.

### 4.2. Characterization of Hydrolysis Products of Commercial Pulps and Their Potential Application

Most of the publications on enzymatic saccharification of raw wood give general time-yield dependences for glucose and reducing sugars [58,59]. Only a few studies have analyzed the monosaccharide composition of hydrolysates [42,46,60], while concentrations of cellobiose, xylobiose, and oligosaccharides are rarely reported [16,17,46]. Finally, data on the content of oligosaccharides with DP $\geq 5$ are practically not found in the literature. However, these data are important characteristic of hydrolysates and their potential in subsequent fermentation using selected bacteria or yeasts. The identification of various oligomers by chromatography is tricky, since their concentrations in hydrolysates against major components are close to the detection limit. Here we propose determination based on the amount of monomers formed from oligomers after acid inversion (Section 2.3). The increase in concentration of the main sugars (glucose, xylose, and mannose) after acid inversion allows for calculating the oligomer content in hydrolysates and their proportion in the total amount of soluble sugars. This approach reveals composition differences in hydrolysates of pulps with similar hydrolyzability.

During hydrolysis by *P. verruculosum* enzymes, kraft pulp hydrolysates accumulated glucose and xylose as major products, while glucose predominated in sulfite pulp hydrolysates (Figure 2). Cellobiose was found to be the second main product of cellulose hydrolysis after glucose (Table 3); an increase in the concentration of xylose and mannose after acid inversion of hydrolysates indicated the presence of soluble oligomeric xylans and mannans. After 48 h of conversion of kraft pulp, the accumulation (up to 4.5 g/L) of xylobiose and xylooligomers with a higher DP was observed, mostly from reprecipitated xylan, highly accessible for *P. verruculosum* endoxylanases. A significant amount of xylooligomers can inhibit β-glucosidase [61], which might explain the observed high cellobiose content in kraft pulp hydrolyzates. The level of pentose oligomers, and especially xylose and xylobiose, must be considered when selecting a fermentation strain for further microbiological conversion. The most complete fermentation of kraft pulp sugar hydrolysates can only be carried out with strains assimilating xylose along with glucose, for example *Bacillus vallismortis* [62], *Candida guilliermondii* FTI 20037 [63], and *B. coagulans* Azu-10 [64]. In addition, to increase the yield of target fermentation products, research is being carried out on the genetic design of *Saccharomyces cerevisiae* [65,66], *Gluconobacter oxydans* [67], and *Lactobacillus plantarum* NCIMB [68].

**Table 3.** Concentrations (g/L) of cellobiose and hemicellulose oligomers in pulp hydrolysates.

| Pulp | Hydrolysis Time | Cellobiose | Monosaccharides from Hemicelluloses Oligomers after Acid Inversion | |
|---|---|---|---|---|
| | | | Xylose | Mannose |
| Sulfite | 24 | 3.7 | 3.4 | 6.5 |
| | 48 | 3.3 | 4.2 | 6.8 |
| Kraft | 24 | 4.2 | 4.6 | 1.7 |
| | 48 | 4.7 | 5.3 | 2.0 |

In the production of easily assimilated hexose sugars, sulfite pulp is more promising over kraft pulp due to the initial softwood composition and cooking method. Sulfite pulp hydrolysates accumulated a significant amount of mannan hydrolysis products, mainly di- and oligomers, since after acid inversion the concentration of mannose increased by 1.7 times. The fermentability of hydrolysates can be improved by increasing the activity of β-mannosidase in the enzyme cocktail; the total amount of easily assimilated hexoses can reach over 93% of the reducing sugars. The high hexose ratio in hydrolysates of wood polysaccharides facilitates microbiological conversion, since most industrial strains preferentially consume hexoses. Yeast strains used in spent sulfite liquor fermentation are known to metabolize in toxic liquor media, assimilating little sugars produced through acid hydrolysis during sulfite cooking [53,66]. For the complete fermentation of media based on enzymatic hydrolysates of sulfite pulp, it is important to use industrial strains capable of utilizing mannose, for example, *Candida* yeasts or genetically modified *Corynebacterium glutamicum* [69] and *Rhodosporidium toruloides-1588* [70].

A large amount (average 10 g/L) of soluble poorly assimilated oligosaccharides in the hydrolyzates of kraft and sulfite pulp provokes the search for their alternative applications. One possible alternative is prebiotics based on cellooligosaccharides (COS), xylooligosaccharides (XOS), and mannooligosaccharides (MOS). These generally indigestible compounds can represent a special type of prebiotics, providing a carbon and energy source for gut microbiota [70,71].

When choosing the best pulp substrate, we should also consider the amount and properties of non-soluble product along with fermentable sugars. Generally, in industrial trials, the insoluble fraction remains significant, which is related to reasonable enzyme dosages and hydrolysis time. Compared to initial polysaccharide composition (Table 1), pulps underwent significant changes after 24 and 48 h of hydrolysis with *P. verruculosum* enzymes (Figure 5). Cellulose remained the main component of unhydrolysed residue; for kraft pulp its content increases to 76%, while for sulfite pulp, after an increase up to 80% (24 h), it then decreased by 1% (48 h). In kraft pulp the hydrolysis rate of xylan, considering its accessibility on the surface of the fibers, can exceed the hydrolysis rate of cellulose. In sulfite pulp xylan content is lower, and its localization significantly decreases its accessibility to xylanases. Furthermore, the content of non-polysaccharide components increased in the sulfite pulp. These two effects contributed to the decrease in the crystallinity of sulfite pulp at the final stages of enzymatic hydrolysis (Figure 4). For kraft pulp, due to intensive hemicelluloses hydrolysis, an enzymatic purification of cellulose took place, increasing both the content of cellulose and its crystallinity.

*4.3. Complex Processing of Industrial Pulps Including Enzymatic Hydrolysis and Microbial Fermentation*

As discussed earlier, spent sulfite liquors undergo biotechnological conversion to obtain bioethanol and single cell protein. The low efficiency of fermentation with yeasts can be increased by increasing the quality of the liquor-based media. Hydrolysates of sulfite pulps obtained with *P. verruculosum* enzymes are glucose-rich solutions with a sugar concentration 3–5 times higher than in the spent liquor. The content of hexoses in hydrolysates can be further increased by increasing the mannanase activity of the enzyme cocktail. Combining even a small amount of hydrolysate with the main stream of spent

sulfite liquor will significantly increase the total hexose content and favorably affect the production of microbial protein or ethanol by yeasts, or various valuable products, if non-yeast strains are used. The unhydrolyzed residue from sulfite pulp contains a mixture of polysaccharides, with cellulose being the major component and hemicelluloses providing good papermaking properties. The addition of this biomodified pulp to the main pulp flow can increase the strength of the resulting paper, ensuring integrated processing of all saccharification products (Figure 6). Thus, this technological solution can preserve a few remained sulfite pulp productions through the introduction of enzymatic hydrolysis and the modernization of the already-applied fermentation processes.

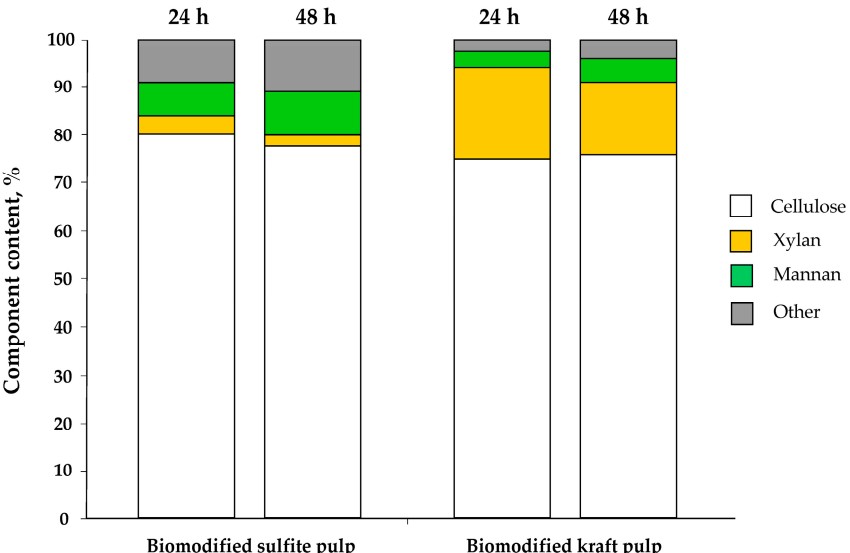

**Figure 5.** Biomodified pulp composition.

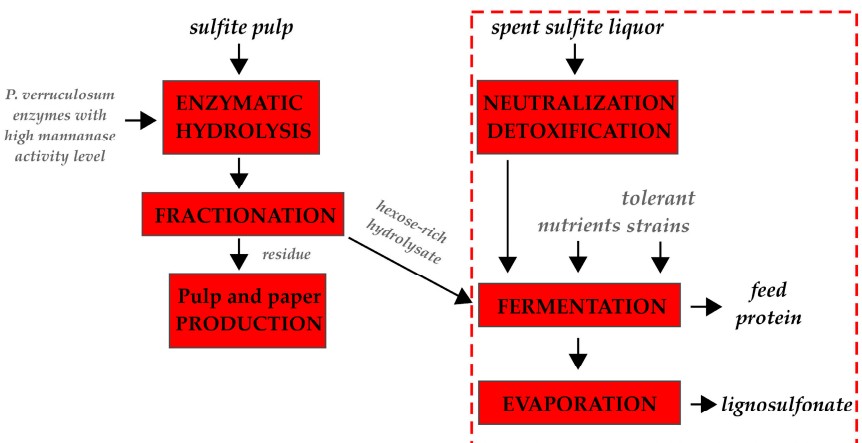

**Figure 6.** Possible strategies of sulfite pulp bioprocessing.

Kraft pulp production has historically lacked fermentation technology. The present work proposes the ways for their implementation, simultaneously increasing the number of products one can obtain from well-studied and widely used commercial kraft pulp. For the enzymatic saccharification of kraft pulp, *P. verruculosum* GHs complexes with sufficiently high xylanase activity can be used without modifications. The resulting solutions containing mostly glucose and xylose should be fermented by industrial strains capable of efficient assimilation of xylose along with glucose. It will allow the most complete conversion of simple sugars into bioproducts for the pharmaceutical, food, or energy industry, such as organic acids, amino acids, and alcohols (including biofuels) (Figure 7). Unfermented XOS and MOS, when skillfully separated from the culture medium after fermentation, can

be used as prebiotics in nutrition, and this topic currently evokes increased interest [72]. Biomodified kraft pulp, an insoluble residue after enzymatic saccharification, has DP and crystallinity close to commercial MCC, obtained by acid hydrolysis of plant cellulose. Further development of enzymatic modification and unhydrolyzed pulp properties will create a background for its applications similar to MCC, including adsorbents, drug carriers, and formulation fillers [73], as well as hydrogel matrix for tissue engineering purposes [27,74].

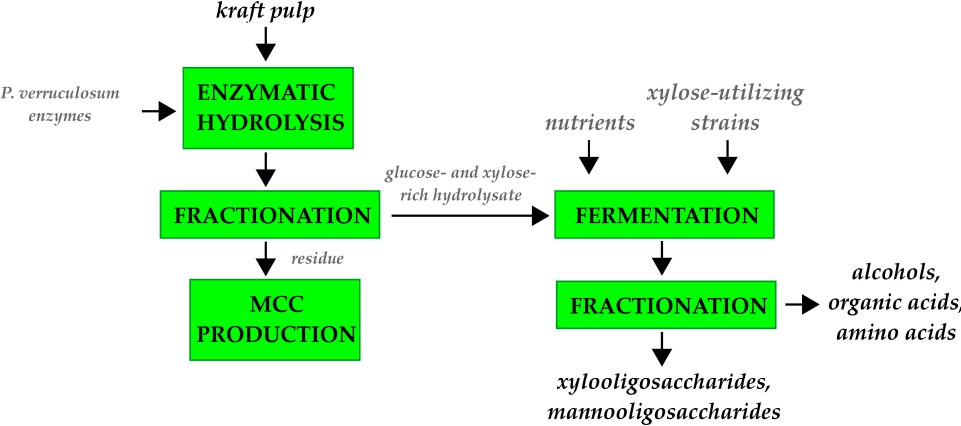

**Figure 7.** Possible strategies for kraft pulp bioprocessing.

## 5. Conclusions

When choosing the best cellulosic substrate for industrial saccharification, it is necessary to consider the specific needs of particular pulp manufacture. For the generation of highly concentrated glucose-rich fermentation media, it is better to use kraft pulp. Advanced and cost-effective kraft pulping allows for heat and reagent recovery, as well as the production of cellulosic pulp suitable for additional applications, in contrast to the direct production of cardboard and paper. A part of the mill's pulp flow can be directed to enzymatic saccharification to produce new products, such as biomodified pulp, which increases biorefining efficiency. While kraft pulp hydrolysis allows for the production of highly concentrated media, the media based on sulfite pulp hydrolysates have enhanced fermentability.

**Supplementary Materials:** The following supporting information can be downloaded at: https://www.mdpi.com/article/10.3390/fermentation9110936/s1, Table S1: Reducing sugar yield and pulp conversion after 48 h of hydrolysis at fiber concentrations of 2.5, 5 and 10%; Figure S1: Scheme of sulfite-pulping process; Figure S2: Scheme of kraft-pulping process; Figure S3: Scheme of pulp bleaching process.

**Author Contributions:** Conceptualization, A.R.S., A.P.S. and A.S.A.; methodology, A.R.S., A.M.R., D.G.C., V.A.P., K.A.M. and A.S.A.; resources, D.A.A. and V.D.T.; data curation, A.R.S., I.N.Z., A.V.M. and E.A.T.; writing—original draft preparation, K.A.M., A.S.A. and A.R.S.; writing—review and editing, D.N.P., M.V.S. and A.P.S.; visualization, D.A.A. and A.S.A.; enzymatic hydrolysis of pulps, A.R.S., K.A.M., M.A.R. and A.S.A.; project administration, A.S.A. All authors have read and agreed to the published version of the manuscript.

**Funding:** The Russian Science Foundation supported all biocatalysis studies, hydrolysate and biomodified pulp analysis (project 22-24-20136).

**Institutional Review Board Statement:** Not applicable.

**Informed Consent Statement:** Not applicable.

**Data Availability Statement:** Not applicable.

**Acknowledgments:** In this work, we used instrumentation of the Core Facility Center "Arktika" of the Northern (Arctic) Federal University named after M.V. Lomonosov and the Federal Research Centre "Fundamentals of Biotechnology" of the Russian Academy of Sciences.

**Conflicts of Interest:** The authors declare no conflict of interest.

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
