# Peer review of "Enzymatic Hydrolysis of Kraft and Sulfite Pulps: What Is the Best Cellulosic Substrate for Industrial Saccharification?"

_fermentation, doi:10.3390/fermentation9110936_

Round 1

Reviewer 1 Report

Comments and Suggestions for Authors

This work has limited value for publication for several reasons: for example: (1) bleached chemical pulps are expensive, why would anybody wants to use bleached chemical pulps for sugar production, it does not make economic sense. (2) If for understanding basioc concepts, it is OK to use commercail pulps, but one would use unbleached pulps, not bleached pulps. (3) One cannot compare softwood pulp with mixed hardwood pulps.  It is known that softwood is harder to saccharify. Ther esults presented here are a;lready well known.

Other issues: Fig. 2 caption, what is the high concentration?  Why sulfute pulp has a higher glucose concentration even with a lower cellulose conversion (Fig. 1).  The reviewer did notice that sulfite pulp has a slightly higher cellulose content. 

Author Response

Dear reviewer!

Thank you very much for your comments and suggestions on our work. Those help us to improve the quality of our manuscript. We are most grateful to you for helping us. Your comments are in italics and our responses are presented below.

 This work has limited value for publication for several reasons, for example:  

  1. Bleached chemical pulps are expensive, why would anybody want to use bleached chemical pulps for sugar production, it does not make economic sense. 

In our work we used bleached sulfite paper-grade pulp, not a dissolving grade pulp. In this way we conducted a comparison of two main industrial methods for delignification and bleaching aimed to produce similar products. We have added our explanation in the text (lines 313-315). Bleached pulps have already been considered as a substrate for enzymatic treatment it the literature (Kafle et al. 2015 (http://dx.doi.org/10.1038/srep15102, new Reference, 18), which confirms the relevance of research (information was added to lines 66-69). In our work we used softwood pulp produced by acid sulfite cooking, its hydrolysability hasn’t been previously described, and compared it to highly available bleaches kraft hardwood paper-grade pulp.

Industrial cooking and bleaching are expensive but also effective ways pre-treat wood to enzymatic saccharification. We discussed here a possibility of using cooking as an effective pretreatment method to saccharify wood substrate. The recalcitrant structure of wood pulp makes any pretreatment method expensive, while finding an effective pretreatment is crucial for wood polysaccharide saccharification. Wood polysaccharides are the so-called second generation of renewable plant substrate which is not a food sources. To reduce costs we proposed to use highly available substrate and commercially available enzymes. A designed scheme, in which existing pulp flow is partially subjected to saccharification, can decrease processing costs and increase the efficacy of sulfite plants.

  1. If for understanding basic concepts, it is OK to use commercial pulps, but one would use unbleached pulps, not bleached pulps. 

Reply: It is well known that residual lignin can significantly decrease enzyme activity. Earlier in our work (Aksenov et al., 2020 (https://doi.org/10.3390/catal10050536) we also reported that bleaching remarkably increased the hydrolysability of commercial pulps.  In addition, the economical sense here lies in the complex processing of bleached pulp to produce paper and also glucose-rich sugar hydrolysates, highly crystalline cellulose residue suitable for high-quality aerogels (Shevchenko et al., https://www.mdpi.com/2073-4344/13/1/103) and probiotic oligosaccharides (line 385-388). We assume that high value of these products can compensate the charges for pulp bleaching.

  1. One cannot compare softwood pulp with mixed hardwood pulps.  It is known that softwood is harder to saccharify. The results presented here are already well known. 

Reply: Thanks for the comment. Earlier in the work of Novozhilov et al., 2016 (https://doi.org/10.1134/S2070050416010098), we showed that softwood pulps can be more easily saccharified compared to hardwood pulps. In this work, we compare two alternative processes, kraft and sulfite pulping, as pretreatment methods before enzymatic hydrolysis by cellulases and hemicellulases. Since hardwoods are not used in the acid sulfite cooking, we used softwood pulp. As reference sample, we used bleached hardwood kraft pulp, which was fundamentally different both in terms of raw materials and production technology. We have shown that two cooking processes have brought these different substrates to the similar level of hydrolysability. Similar results have not been previously described in the literature. The novelty of the work is also related to new commercially available P. verruculosum enzymes with high xylanase activity and declared application in feed industry. However, their saccharifying effect on pulp substrates turned out to be very good, which is extremely important from a practical point of view.

 Other issues: Fig. 2 caption, what is the high concentration?  Why sulfite pulp has a higher glucose concentration even with a lower cellulose conversion (Fig. 1).  The reviewer did notice that sulfite pulp has a slightly higher cellulose content.  

Reply: We referred 10% pulp concentration as “high” (comparing to 2.5 and 5% which are referred as “low”). We have modified the Fig.2 caption.  When comparing results presented in Fig. 1 and 2 it should be noted that higher pulp concentration lead to lower cellulose conversion. For sulfite pulp, cellulose conversion decreased from 53 to 42% when pulp concentration increased from 5 to 10 % (mostly due to stirring difficulties). For kraft pulp, with increased pulp concentration the amount of available xylan on the surface of the fibers also increased, which can lead to more severe inhibition of P. verruculosum enzymes with xylobiose (ref. 60 Wang et al., https://doi.org/10.1021/acs.energyfuels.8b01424). In our opinion, it can contribute to similar glucose yield for two substrates. Cellulose content initially is indeed higher in sulfite pulp, which had positive effect on a final glucose concentration in sulfite pulp hydrolysates. However, sulfite pulp contains more impurities, which affect the properties of unhydrolysed residue, such as crystallinity.

Kind regards,  

Andrey Aksenov on behalf of the authors 

Reviewer 2 Report

Comments and Suggestions for Authors

1. Line 55-56, the reason should be added.

2. Line 63-68, the latest reference should be added. You can check the work in https://doi.org/10.3390/fermentation8100558

3. In the introduction, the background of kraft pulping should also be introduced

4. Line 176-177, why two  fiber concentrations of 2.5 and 5.0% was investigated. It should be explained.

5. Line 231, how to understand the “biomodification”? I think it is the incorrect description.

6. Line 245, the information of  negative factors should be stated.

7. Line 386-290, the exited lignin in the pulp can also affect the high accessibility of pulps to enzymatic hydrolysis. It should be discussed. You can check the works in https://doi.org/10.1016/j.rser.2021.111822

8. Line 417-418, the reference should be added. You can check the work in https://doi.org/10.3390/ijms241713422

9. The size of the all figures should be Uniformed, as some of them too big. 

Comments on the Quality of English Language

no

Author Response

Dear reviewer!

We are very grateful for the work you have done with our paper. We tried to take into account all your recommendations. Please find below our point-by-point itemized answers and corrections. We hope that the manuscript’s revised version with corrections and additions meets the acceptable quality for publication in the journal

  1. Line 55-56, the reason should be added.

Reply: We corrected the text as suggested, lines 54-57

  1. Line 63-68, the latest reference should be added. You can check the work in https://doi.org/10.3390/fermentation8100558.

Reply: Corrected as suggested (ref. 20)

  1. In the introduction, the background of kraft pulping should also be introduced.

Reply:  Corrected as suggested, lines 62-66

  1. Line 176-177, why two fiber concentrations of 2.5 and 5.0% was investigated. It should be explained.

Reply: Thank you for the important recommendation. The efficacy of hydrolysis decreases with increase in pulp concentration. It is mainly due to increased concentration of sugars and inhibition by product (Zhang X. et al. 2009 doi: 10.1016/j.biortech.2009.06.082), as well as stirring issues. At the same time, too low pulp concentrations are economically ineffective and lead to low sugar content in hydrolysates. We chose 2.5 % pulp concentration to ensure complete hydrolysis; this concentration was applied in industrial acid hydrolysis to obtain media with fermentable sugar content of less than 3 % [Rabinovich, M. L. Wood hydrolysis industry in the Soviet Union and Russia: a minireview. Cell. Chem. Tech. 2010.]. And we used 5 % concentration since we have previously shown that kraft pulp could be still efficiently hydrolyzed with P. verruculosum enzymes at this concentration [Aksenov et al. 2020 doi.org/10.3390/catal10050536]. (At “high concentration” of 10 % used in the study hydrolysis is incomplete)

  1. Line 231, how to understand the “biomodification”? I think it is the incorrect description.

Reply: We have corrected the sentence, line 247-248.

  1. Line 245, the information of negative factors should be stated.

Reply: We discussed negative factors in lines 195-197, here we meant the negative effect of residual lignin in particularly

  1. Line 286-290, the exited lignin in the pulp can also affect the high accessibility of pulps to enzymatic hydrolysis. It should be discussed. You can check the works in https://doi.org/10.1016/j.rser.2021.111822

Reply:  Corrected as suggested, ref. 41

  1. Line 417-418, the reference should be added. You can check the work in https://doi.org/10.3390/ijms241713422

Reply: Corrected as suggested, lines 434-436.

  1. The size of the all figures should be Uniformed, as some of them too big.

Reply: Corrected as suggested

Kind regards,

Andrey Aksenov on behalf of the authors

Reviewer 3 Report

Comments and Suggestions for Authors

Cellulose biomass resources represent a highly promising source of industrial fermentation raw materials, with their hydrolysis pretreatment being one of the key technological challenges. Despite the availability of efficient industrial methods for cellulose and hemicellulose hydrolysis, the hydrolysis and utilization of lignin remain exceedingly challenging. Current work compares chemistry and technological features of two different cooking processes in the preparation of polysaccharide substrates for deep saccharification with P. verruculosum glycosyl hydrolases. The work is interesting and provides novel insights on the mechanism of enzyme hydrolysis of cellulose biomass. The results also have the potential for practical use in industry. For the most part, this work is based on sound designs and experiments.  

It is important to include a detailed description of the statistical analysis methods in the Materials and Methods section. Additionally, in the Results section, you should incorporate p-values and conduct a thorough significance analysis to robustly support your conclusions. These revisions will enhance the overall quality and scientific rigor of your research, aligning it with the standards expected by prestigious journals in the field of biology.

Manuscript can be further improved taking following points into consideration

Comments

1. L116 unit should be supplemented in Table 1.

2. L126 Table 2, U also should be presented in the table body, not in table.

3. How do the authors define "low pulp concentrations," and what criteria were used to establish the different concentration conditions?

4. The legend of Figure 2 is confusing.

5. Can the authors provide additional data on fermentation inhibitory substances such as weak acids, furan compounds, and phenolic compounds produced during the hydrolysis of cellulose biomass, which are important indicators for assessing the effectiveness of hydrolysis processes? This data could be used for comparisons between different processes.

6. Most of the content in the discussion should be moved to the results section.

Comments on the Quality of English Language

no

Author Response

Dear reviewer!

We are very grateful for the work you have done with our paper We tried to take into account your recommendations and we hope that we  have raised the level of our article. Please find below our point-by-point itemized answers and corrections.

It is important to include a detailed description of the statistical analysis methods in the Materials and Methods section. Additionally, in the Results section, you should incorporate p-values and conduct a thorough significance analysis to robustly support your conclusions.

Reply: Thanks for the remark. We have corrected the Method section (lines 141-142). In our work we used triplicates and standard deviation, and didn’t use p-values for result comparison. 

  1. L116 unit should be supplemented in Table 1.

Reply: We have corrected the table, line 127.

  1. L126 Table 2, U also should be presented in the table body, not in table.

Reply: It is the same units, “U per 10 FPU”, for all table columns, we found it excessive to duplicate the same unit in each column.

  1. How do the authors define "low pulp concentrations," and what criteria were used to establish the different concentration conditions?

Reply: We referred pulp concentration of 2.5 and 5 % as “low” comparing to “high” concentration of 10 %. Industrial production of ethanol from plant sources applies plant mass concentration of up to 10 %. Higher substrate concentrations of 10-30 % are applied in starch-based enzymatic hydrolysis. High concentration of sugars in resulting hydrolysates is crucial for subsequent biosynthesis of organic acids and amino acids. However, in the case of wood pulp substrates increasing of pulp concentration (and subsequently sugar concentration) is limited by hydrolysis efficiency, the optimization of pretreatment and hydrolysis methods is still relevant [Zhang X. et al. 2009 doi: 10.1016/j.biortech.2009.06.082, Wu J. et al. 2023 doi: 10.1016/j.biortech.2023.128647].

  1. The legend of Figure 2 is confusing.

Reply: We divided Fig. 2 into two parts to simplify the legend.

  1. Can the authors provide additional data on fermentation inhibitory substances such as weak acids, furan compounds, and phenolic compounds produced during the hydrolysis of cellulose biomass, which are important indicators for assessing the effectiveness of hydrolysis processes? This data could be used for comparisons between different processes.

Reply:  In contrast to acid hydrolysis, enzymatic hydrolysis of wood pulps does not produce any harmful compounds affecting subsequent fermentation [Caoxing Huang et al. 2022 https://doi.org/10.1016/j.rser.2021.111822]. Earlier, we have successfully fermented kraft pulp hydrolyates into organic acids and amino acids (ref. in Russian)

(https://scholar.google.com/scholarhl=ru&as_sdt=0%2C5&q=ДВУХСТАДИЙНАЯ+БИОКОНВЕРСИЯ+ПОЛИСАХАРИДОВ+ДРЕВЕСНОГО+ПРОИСХОЖДЕНИЯ+В+ПРОДУКТЫ+С+ВЫСОКОЙ+ДОБАВЛЕННОЙ+СТОИМОСТЬЮ&btnG=,

https://scholar.google.com/scholarhl=ru&as_sdt=0%2C5&q=КОМПЛЕКСНОЕ+БИОТЕХНОЛОГИЧЕСКОЕ+ИСПОЛЬЗОВАНИЕ+ЛИСТВЕННОЙ+ЦЕЛЛЮЛОЗЫ+С+ПРИМЕНЕНИЕМ+ПРЕПАРАТОВ+КАРБОГИДРАЗ+И+BACILLUS+COAGULANS&btnG= )

  1. Most of the content in the discussion should be moved to the results section.

Reply: We added text to the Results section and corrected Discussion section as well.

Kind regards,

Andrey Aksenov on behalf of the authors

Round 2

Reviewer 2 Report

Comments and Suggestions for Authors

The authors have revised the manuscript according to the reviewer. It can be accepted now

Comments on the Quality of English Language

The authors have revised the manuscript according to the reviewer. It can be accepted now

Author Response

Dear reviewer!

Thank you very much for your comments and suggestions on our work! Those help us to improve the quality of our manuscript. We are most grateful to you for helping us.